# Beyond the convexity assumption: Realistic tabular data generation under quantifier-free real linear constraints

**Mihaela Cătălina Stoian**[1]                    MIHAELA.STOIAN@ST-HILDAS.OX.AC.UK
**Eleonora Giunchiglia**[2]                    E.GIUNCHIGLIA@IMPERIAL.AC.UK
[1] *University of Oxford,* [2] *Imperial College London*

**Editors:** Leilani H. Gilpin, Eleonora Giunchiglia, Pascal Hitzler, and Emile van Krieken

## Abstract

Synthetic tabular data generation is a challenging problem due to the high complexity of the underlying distributions that characterise this type of data. Despite recent advances in deep generative models (DGMs), existing methods often fail to produce realistic datapoints that are well-aligned with available background knowledge. In this paper, we address this limitation by introducing Disjunctive Refinement Layer (DRL), a novel layer designed to enforce the alignment of generated data with the background knowledge specified in user-defined constraints. DRL is the first method able to automatically make deep learning models inherently compliant with constraints as expressive as quantifier-free linear formulas, which can define non-convex and even disconnected spaces. Our experimental analysis shows that DRL not only guarantees constraint satisfaction but also improves efficacy in downstream tasks. Indeed, it improves performance metrics by up to 21.4% in F1-score and 20.9% in Area Under the ROC Curve.

## 1. Main Contributions

In this paper, we propose a novel layer—called Disjunctive Refinement Layer (DRL)—able to constrain any DGM output space according to background knowledge expressed as Quantifier-Free Linear Real Arithmetic (QFLRA) formulas. QFLRA formulas can capture any relationship over the features that can be represented as a combination of conjunctions, disjunctions and negations of linear inequalities. Thanks to their expressivity, QFLRA formulas can define spaces that are not only non-convex but can also be disconnected. On the contrary, linear inequalities can only capture convex output spaces.

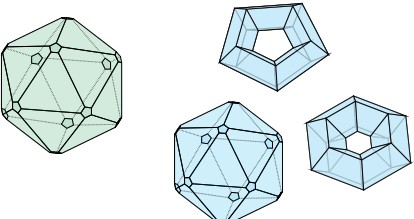

Figure 1: Example of spaces defined by (green) a set of linear inequalities and (blue) a set of QFLRA formulas.

See Figure 1 for an example of spaces defined by linear inequalities and QFLRA formulas. While linear inequalities establish a single lower and upper bound (if existent) for each feature, QFLRA formulas define multiple intervals where the background knowledge holds, each with its own boundaries, thus significantly increasing the complexity of the problem.

**Example 1** *The knowledge: "The value of $x_5$ should be always at least $x_1$, and if greater than $x_2$ then it should also be at least equal to $x_3$. Also, $x_5$ should never be greater than $x_4$", that cannot be expressed by a set of linear inequalities, corresponds to the QFLRA formula:*

$$(x_5 \geq x_1) \wedge ((x_5 > x_2) \rightarrow (x_5 \geq x_3)) \wedge (x_5 \leq x_4). \tag{1}$$

*Moreover, this formula entails other hidden relations such as, e.g., $\neg(x_1 > x_4)$.*

Table 1: DGMs vs. DGMs+DRL on classification datasets.

| | F1 | | | | wF1 | | | | AUC | | | |
|---|---|---|---|---|---|---|---|---|---|---|---|---|
| | URL | CCS | LCLD | Heloc | URL | CCS | LCLD | Heloc | URL | CCS | LCLD | Heloc |
| WGAN | 0.794 | 0.303 | 0.139 | 0.665 | 0.796 | 0.330 | 0.296 | 0.648 | 0.870 | 0.814 | 0.605 | **0.717** |
| + DRL | **0.800** | **0.313** | **0.197** | **0.721** | **0.801** | **0.340** | **0.339** | **0.652** | **0.875** | **0.885** | **0.623** | 0.717 |
| TableGAN | 0.562 | **0.196** | 0.259 | 0.593 | 0.659 | **0.228** | 0.393 | 0.615 | 0.843 | **0.802** | 0.655 | 0.707 |
| + DRL | **0.619** | 0.163 | **0.269** | **0.628** | **0.693** | 0.196 | **0.401** | **0.628** | **0.865** | 0.742 | **0.657** | **0.709** |
| CTGAN | 0.822 | 0.145 | 0.247 | 0.736 | 0.799 | 0.159 | 0.379 | 0.675 | 0.859 | 0.914 | **0.651** | 0.744 |
| + DRL | **0.836** | **0.288** | **0.288** | **0.744** | **0.815** | **0.308** | **0.409** | **0.680** | **0.883** | **0.955** | 0.643 | **0.745** |
| TVAE | 0.810 | 0.325 | 0.185 | 0.717 | 0.802 | 0.351 | **0.330** | 0.686 | 0.863 | 0.858 | 0.631 | 0.750 |
| + DRL | **0.835** | **0.467** | **0.189** | **0.731** | **0.832** | **0.487** | **0.330** | **0.694** | **0.893** | **0.926** | **0.635** | **0.752** |
| GOGGLE | 0.622 | 0.039 | 0.248 | 0.596 | 0.648 | 0.076 | 0.296 | 0.566 | 0.742 | 0.549 | 0.551 | 0.600 |
| + DRL | **0.720** | **0.253** | **0.298** | **0.698** | **0.673** | **0.281** | 0.310 | **0.636** | **0.747** | **0.758** | **0.563** | **0.691** |

To derive such additional hidden relations, we developed a novel variable elimination method which generalises the analogous procedure for systems of linear inequalities based on the Fourier-Motzkin result (see, e.g., (Dechter, 1999)). Once compiled, by definition, DRL (i) guarantees the satisfaction of the constraints, (ii) can be seamlessly added to the topology of any neural model, (iii) allows the backpropagation of the gradients at training time, (iv) performs all the computations in a single forward pass (i.e., no cycles), and (v) given a sample generated by a DGM, it returns a new one that is optimal with respect to the original (intuitively, which minimally differs from the original sample while taking into account the user preferences on which features should be changed first).

## 2. Experimental Analysis Results

We tested our method on the challenging tabular data generation task. We considered five datasets: URL, CCS, LCLD, and Helocare used for classification tasks, while Houseis used for regression. In Table 2 we report the *constraint violation rate* (CVR) on these datasets using five state-of-the-art genera-

Table 2: CVR for each model and dataset. Cases with CVR≥50% are underlined. Best results are in bold.

| | URL | CCS | LCLD | Heloc | House |
|---|---|---|---|---|---|
| WGAN | 22.8±4.9 | 44.7±7.1 | 47.5±14.5 | 80.6±9.3 | 100.0±0.0 |
| TableGAN | 8.5±2.2 | 61.2±13.3 | 32.0±4.7 | 59.9±16.7 | 100.0±0.0 |
| CTGAN | 9.7±2.0 | 78.5±5.7 | 7.1±1.3 | 56.6±9.8 | 100.0±0.0 |
| TVAE | 10.3±1.1 | 16.9±1.6 | 10.3±0.6 | 44.9±1.0 | 100.0±0.0 |
| GOGGLE | 7.3±8.1 | 60.3±6.8 | 70.4±16.1 | 52.7±6.3 | 100.0±0.0 |
| All + DRL | **0.0±0.0** | **0.0±0.0** | **0.0±0.0** | **0.0±0.0** | **0.0±0.0** |

tive models: WGAN (Arjovsky et al., 2017), TableGAN (Park et al., 2018), CTGAN (Xu et al., 2019), TVAE (Xu et al., 2019), and GOGGLE (Liu et al., 2022). The last line reports the CVR obtained by adding DRL to each of the models. In Table 1 we report the efficacy of the standard DGMs, and the DGMs trained with DRL. The Table clearly shows that adding DRL during training and inference not only makes the synthetic data compliant by-design with the constraints, but also it improves the efficacy of the models.

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
