# OpenReview forum: "Beyond the convexity assumption: Realistic tabular data generation under quantifier-free real linear constraints"
_nesyconf.org/NeSy/2025/Conference_Phase_2 — NeSy 2025 - Phase 2 Poster_

### Official Review · Reviewer_pZ6z · 2025-07-05
**Beyond the convexity assumption: Realistic tabular data generation under quantifier-free real linear constraints**

**Rating:** 6
**Confidence:** 4

**Review:**

The extended abstract "Beyond the convexity assumption: Realistic tabular data generation under quantifier-free real linear constraints" summarizes a paper with the same title published at the "Thirteenth International Conference on Learning Representations" (ICLR 2025). The paper was presented as a poster at ICLR 2025 and is related to neurosymbolic reasoning and learning, although rather special aspects of neurosymbolic integration are considered, in particular, the generation of synthetic tabular data, the extension from constraints as linear inequalities to constraints as quantifier-free linear real arithmetic formulas etc. The extended abstract contains the two major evaluation tables from the long paper.

**Anonymity:**

Remain anonymous

---

### Official Review · Reviewer_wfzK · 2025-07-05
**Promising work, very appropriate to the conference**

**Rating:** 7
**Confidence:** 3

**Review:**

The authors provide an extended abstract for an innovative method of generating synthetic data while respecting the inherent constraints of the data being generated. The method consists of a differentiable layer that can be applied to deep generative models to return a constraint-compliant data sample, making it quite general and easy to apply to existing solutions.
The abstract is in line with the conference as it uses symbolic background knowledge to augment DL methods in a hard task like synthetic data generation, where purely DL methods seemingly have been the go-to.
I would be interested to understand more clearly how the background knowledge is derived and how hard this step is, as it seems like a crucial bottleneck of the method and, as I understand, the constraints must be user-defined. Regardless, the results seem promising while the method seems very sound and general. For this, I would recommend the extended abstract to be accepted to the conference.

**Anonymity:**

Remain anonymous